# Comparative Assessment of Phytoconstituents, Antioxidant Activity and Chemical Analysis of Different Parts of Milk Thistle *Silybum marianum* L.

**DOI:** 10.3390/molecules27092641

**Published:** 2022-04-20

**Authors:** Ansar Javeed, Maqsood Ahmed, Allah Rakha Sajid, Aatika Sikandar, Muhammad Aslam, Talfoor ul Hassan, Samiullah Dogar, Zahid Nazir, Mingshan Ji, Cong Li

**Affiliations:** 1College of Bioscience and Biotechnology, Shenyang Agricultural University, Shenyang 110866, China; ansarjaveed@henu.edu.cn; 2School of Life Sciences, Henan University, Kaifeng 475000, China; 3College of Plant Protection, Shenyang Agricultural University, Shenyang 110866, China; maqsoodahmed200@hotmail.com (M.A.); aatika_sikander@yahoo.com (A.S.); jimingshan@syau.edu.cn (M.J.); 4Agriculture Department (Plant Protection) Pest Warning & Quality Control of Pesticides, Gujrat 50700, Pakistan; samiullahdogar42@gmail.com; 5Directorate General of Pest Warning and Quality Control of Pesticides, Lahore 42000, Pakistan; allahrakhasajid@yahoo.com (A.R.S.); draslamdir@gmail.com (M.A.); ddappgujranwala@gmail.com (T.u.H.); drzahidnazir@gmail.com (Z.N.)

**Keywords:** Milk thistle, silymarin, phytochemicals, antioxidant activity, GC-MS

## Abstract

*Silybum marianum* L. is a therapeutic plant belonging to the family Asteraceae, which has exhibited silymarin, a principal component used to cure various physiochemical disorders. The study appraised the phytochemical analysis, antioxidant activity and chemical analysis of an extract from the seed, stem and leaves. Qualitative and quantitative phytochemical analysis was evaluated by the Folin–Ciocalteu reagent method and aluminum chloride colorimetric method, respectively. While the antioxidant activity was determined by using 1,1-diphenyl-2-picrylhydrazyl (DPPH) and acetate buffer in ferric chloride (FRAP) assay, respectively, the chemical profile was evaluated by Gas Chromatography-Mass Spectrometry (GC-MS) assay. The study outcomes identified that alkaloids, glycosides, flavonoids, terpenoids, steroids and catcholic tannins were present in seed, stem and leaves extracts except for saponins and Gallic tannins. Whereas, phenols were absent only in seed extract. Quantitative analysis revealed the presence of phenols and flavonoids in appreciable amounts of 21.79 (GAE/g), 129.66 (QE/g) and 17.29 (GAE/g), 114.29 (QE/g) from the leaves and stem extract, respectively. Similarly, all extracts expressed reasonable DPPH inhibition (IC_50_) and FRAP reducing power such as 75.98, 72.39 and 63.21% and 46.60, 51.40 and 41.30 mmol/g from the seeds, stem and leaves extract, respectively. Additionally, chemical analysis revealed the existence of 6, 8 and 9 chemical compounds from the seeds, stem and leaves extract, respectively, corresponding to 99.95, 99.96 and 98.89% of the whole extract. The chemical compound, Dibutyl phthalate was reported from all extracts while, Hexadecanoic acid, methyl ester and Silane, (1,1-dimethylethyl), dimethyl (phenylmethoxy) were reported only from the seed and leaves extract. Moreover, Methyl stearate was also a major compound reported from all extracts except for seed extract. It is demonstrable that extracts from different parts of *S. marianum* possess significant antioxidant activity, as well as valuable chemical compounds accountable for therapeutic effects that might be incorporated as an alternative to synthetic chemical agents.

## 1. Introduction

Herbs and shrubs are an important source for the treatment of several disorders because of the presence of numerous phytochemicals. Among these natural plant species, Milk thistle (*Silybum marianum* L.) is considered an important and ancient therapeutic plant. It is native to southern areas of Europe, Northern Africa, South and North America, and Australia, as well as in some parts of Asia where it is used to cure liver diseases and is beneficial for lactating mothers [1,2,3]. It has been cultivated extensively in certain parts of India, while it grows naturally as a weed in Pakistan [4]. The main reason for the production of *S. marianum* is the presence of a chemical component known as silymarin, which is a vital component used as a curing agent for the liver and other physical disorders and hence it earned the status as a liver tonic in the current scenario.

Essential materials produced by the plants are secondary metabolites, which protect them against various pathogens [5]. Despite the valuable essential oils and pharmacological properties, *S. marianum* has gained limited attention in respect of its chemical composition. Chemical profiling and antioxidants activities, as well as antimicrobial activities of flower extracts of Moroccan chamomile (*Cladanthus mixtus*) were reported to be dependent upon numerous factors like environmental conditions, genetic variances and the plant part used for assessment [6]. In addition, other parameters, such as harvest season, parts used and extraction procedures, extensively affect the extract yield, their composition and, ultimately, their bioactivity [7].

*Silybum marianum* is used for medicinal purposes for the treatment of various diseases due to the presence of antioxidants, total phenolics and flavonoids. Studies have reported the biological activities of silymarin seed oil such as its ability to scavenge free radicals, and the fact that it contains linoleic (omega-6) and oleic (omega-9) acids involved in preventing atherosclerosis, diabetes, and cancer, lung and hepatocellular cancers [8,9,10,11,12,13]. Some of the other active components exhibited by *S. marianum* seeds are silybonol, apigenin, betaine, proteins and free fatty acids [14]. Moreover, the oil obtained from Milk thistle seed has also been suggested as a rich source of vitamin E and a probable source of natural antioxidants and this antioxidant activity is significantly higher than artificially prepared flavonoid/flavono-lignans [15,16]. Proximate analysis of *S. marianum* seeds comprised of oil content, moisture content, ash content, crude fiber, carbohydrates content, and total proteins 26.05, 4.48, 1.93, 5.48, 87.2, 23%, respectively [4,17]. However, silymarin contents in *S. marianum* plants may differ according to varieties, geographical areas and climatic conditions where it grows [18].

Silymarin, a major constituent is mostly contained in the leaves, roots, seeds and fruit of this plant [19,20]. However, the ripped seeds of *S. marianum* contain relatively more silymarin, which is also involved in the regeneration of spoiled hepatic tissues [8,21]. Silybin, a core part of silymarin, inhibits oxidative stress, reduces fat accumulation in the human liver and insulin resistance [22,23]. Clinically, Silybin also possessed anti-Parkinson and anti-Alzheimer properties [24,25]. The secondary metabolites are the main components of the natural plant extracts existing as phenols and flavonoids which act as antioxidant agents [26]. Likewise, *S. marianum* is a vital plant in lieu of its significant antioxidant properties [27]. Moreover, natural antioxidants are becoming a core area of interest in advanced scientific exploration [28,29]. These natural antioxidants exhibited by plants contain anti-microbial, anti-viral, anti-allergic, anti-inflammatory, anti-mutagenic, anti-cancer and anti-aging properties [30,31].

Similarly, extraction methods affect the yield and phytoconstituents qualitatively and quantitatively. The best records of yield, total phenolics, and flavonoids contents from *Cladanthus mixtus* were obtained via Soxhlet extraction methods while using methanol as solvent. Among extracts, methanol was recorded as presenting the highest antiradical activity IC_50_ = 55.50 µg/mL, while the non-polar solvent, n-hexane showed the lowest antiradical activity IC_50_ = 259 µg/mL [32]. Moreover, for nutrient and amino acid screening from the essential oil of *Origanum compactum* different techniques like microwave-assisted extraction (MW) and Clevenger hydrodistillation (HD) were used [33]. Chaqroune and Taleb [34] evaluated the effects of solvent and extraction techniques on total phenolics, flavonoids content and antioxidant activities from the aerial parts of the rosemary plant and reported that all tested factors significantly impact yield, total phenolics, flavonoids contents and antioxidant activities. To the best of our knowledge, no detailed data about phytochemicals constituents and concerned biological activities in different parts of *S. marianum* wildly growing in Pakistan is published. Because of the diverse biological activities and extensive usage of this plant as a therapeutic perspective, the current research is performed with the objectives to assess the qualitative and quantitative phytochemical analysis, to assess antioxidant activities of different parts and to compare extracts yield obtained from different parts of *S. marianum.*

## 2. Materials and Methods

### 2.1. Plant Material

Samples of the plant (leaves, stem and roots) were collected from District Sialkot Punjab, Pakistan during the months of June–July 2019 with longitudinal gradients 32.66843 N 32°40′6.348″ 74.17298 E 74°10′22.734″ with a humid subtropical climate under the Koppen climate classification with four seasons. Over the course of the year, temperatures varied from 42 °F to 103 °F.

Firstly, the plant species has been botanically authenticated by Dr. Dilbar Hussain, Ayub Agriculture Research Institute (AARI) Entomological Research Institute, Faisalabad Pakistan. All the samples (leaves, stem and roots) were collected from the same plant at the mature/full blooming stage of the plants. Caution was taken that only healthy plants were selected for sample collection.

### 2.2. Preparation of Extract

The collected samples were dried under shade for 20 days at room temperature to avoid photo-oxidation and ground to a fine powder using an electric blender, then the dried powder was extracted for 72 h using methanol (MeOH) 95% by cold extraction method at room temperature.

Being a polar solvent, methanol has good penetration ability to the cell content and exhibited the capacity to extract both the polar and non-polar compounds and to dissolve primary and secondary metabolites. Moreover, methanol boils at 64.7 °C and the temperature needs to be lowered to evaporate the solvent which in return less damages the extract. For extraction, 10 g of each dried sample was extracted with 100 mL of methanol 95% separately with a sample to solvent ratio (10:100) [35,36]. Three replications were performed for each sample extraction. Extracted contents were filtered using Whatman filter paper No. 1 and concentrated on a rotary evaporator (model R-210 BUCHI Labortechnik AG, CH-9230 Flawil 1/Switzerland) to reduce the volume. On complete drying of the solvent using a fume cupboard for 24 h; the obtained yield was measured by applying Equation (1). The obtained samples were stored at 4 °C in glass stopper vials to study their phytochemical analysis and antioxidant activities [37].
(1)$$\mathrm{Yield}\;\left(\%\right)=\frac{\;\mathrm{Weight}\;\mathrm{of}\;\mathrm{the}\;\mathrm{extract}\;\left(g\right)}{\;\mathrm{Weight}\;\mathrm{of}\;\mathrm{the}\;\mathrm{dried}\;\mathrm{sample}\;\left(g\right)\;}\times 100\%$$

### 2.3. Qualitative Phytochemical Analysis

The qualitative phytochemical analysis of the extracts was validated using standard protocols to analyze the existence of saponins, alkaloids, steroids and terpenoids [38], glycosides, flavonoids, and flavones [39,40], Phenols [41] and tannins [42].

### 2.4. Quantitative Analysis

#### 2.4.1. Analysis for Total Phenolic Contents

Quantitative analysis for phenols was assessed by the Folin–Ciocalteu reagent method. In Brief, 1 mg of extract was dissolved in 1 mL methanol. The resulting mixture was shaken vigorously and separated for analysis. For the determination of phenols, Folin–Ciocalteu reagent assay was used. A total of 1 mL solution (1 mg mL^−1^) was poured into 2.5 mL of Folin–Ciocalteu reagent (10%) along with 2 mL solution of (2%) sodium carbonate (Na_2_CO_3_). Then following consequent mixture was incubated in the darkness for 15 min to measure absorbance at 765 nm in a 96-well ELISA plate using (Spectra-Max 190, Meigu Molecular International Co., Ltd., Shanghai, China). Gallic acid was used as 1 mg mL^−1^ to build a standard curve and following concentrations 1, 0.50, 0.25, 0.10, 0.05, 0.02, 0.01, 0 mg mL^−1^ were used to designate the obtained results as equivalent of Gallic acid (GAE) mgg^−1^ of total extract [43,44]. All the treatments were replicated ten times to analyze the data.

#### 2.4.2. Analysis of Total Flavonoids Content

On the other hand, contents for flavonoids were assessed by the aluminum chloride colorimetric method. In brief, 1 mL solution (1 mg mL^−1^) was added into 3 mL methanol, with subsequent addition of (10%) aluminum chloride (AlCl_3_) 0.2 mL, 1 M potassium acetate 0.2 mL, and purified water amounting to 5.6 mL. The resulting mixture was incubated in the darkness for 30 min to measure the absorbance against the prepared reagent blank (without adding extract) at 420 nm. However, the blank was executed using distilled water. Quercetin was used as a standard at different concentrations (1, 0.50, 0.25, 0.10, 0.05, 0.02, 0.01, 0 mg mL^−1^) to build a standard curve as quercetin equivalent (QE) mgg^−1^ of total extract [43,45]. Each treatment was replicated 10 times.

### 2.5. Assessment of Free Radical Scavenging Activity DPPH Assay

DPPH free radical scavenging activity was validated to screen the antioxidant activity of plant extracts. To evaluate the free radical scavenging/antioxidant activity of the extracts, free radical 1,1-diphenyl-2-picrylhydrazyl (DPPH) (C_18_H_13_N_5_O_6_) was used in a 1% solution of tween 20 [46]. Briefly, the solution of DPPH was prepared in HPLC grade methanol (0.004 g 100 mL^−1^). Then, into freshly prepared 3.5 mL DPPH solution, 0.25 mL extract solution prepared in methanol separately for each plant part was added. The consequent mixture was shaken for complete mixing and then incubated in darkness for 30 min at 28 °C [47]. After incubation, the absorbance at 517 nm was assessed by (Spectra-Max 190, Meigu Molecular International Co., Ltd., Shanghai, China). Methanol was used as a negative control, whereas DPPH and methanol were used for experimental control. The inhibition percentage of primed DPPH solution was predicted on the reduction in absorbance through Equation (2). Values of lower absorbance represent higher free radical scavenging activity [48].
(2)$$\mathrm{Inhibition}\;\left({\mathrm{IC}}_{50}\right)=\frac{A_{blank}-A_{sample}}{A_{blank}}\times 100$$
where: *A_blank_* = (control absorbance); *A_Sample_* = (samples absorbance).

### 2.6. Ferric Reducing Antioxidant Power FRAP Assay

For the determination of FRAP in different parts of *S. marianum*, a modified method proposed by Benzie and Strain [49] was used. The employed reagent for this purpose included 300 mM acetate buffer exhibited (pH 3.6), 10 mM TPTZ in 40 mM Hydrochloric acid (HCl) and 20 mM Ferric Chloride (FeCl_3_) + 6H_2_O in purified water. The consequent mixtures were mixed together with the ratio of 10:1:1. Next, the resulting solution was heated at 37 °C using a water bath prior to use. Extracts from various parts of *S. marianum* at concentrations (50 µg mL^−1^) were permitted to react in the darkness for half an hour with 3.0 mL of FRAP reagent. The absorbance at 593 nm was assessed by (Spectra-Max 190, Meigu Molecular International Co. Ltd., Shanghai, China). To construct a standard curve FeSO_4_ (100–1500 μM) was used as a standard; whereas ascorbic acid and Butylated hydroxytoluene (BHT) were employed as positive controls and in the last, results were described as mmol Fe (II)/g of dry extract.

### 2.7. Chemical Analysis

Chemical analysis using GC-MS was accomplished by AGILENT 6890-5973N accomplished with a gas chromatograph established on HP1 capillary column Model Number: TG-5MS (30 m × 250 μm × 0.25 μm) with the flow: 1.0 mL/min. Other fixed parameters were: starting temperature of the oven was 70 °C for 0 min and final temperature (220 °C) for final time (10 min). Whereas, the inlet temperature of 250 °C was adjusted with a split ratio of 10:1. Other heat parameters were; Thermal Aux temp 285 °C, with 35–520 units of MS scan range, MS source temperature 230 °C, MS quadruple temperatures 150 °C units; however, helium was used as a carrier with stream frequency of 1.0 mL min^−1^. HPLC grade methanol was used as blank. The prevailing compounds in the extract were authenticated and confirmed by linking with GC-MS literature data provided by the National Institute of Standards and Technology Mass Spectral database Wiley/NIST.1998.1 [48]. 

### 2.8. Statistical Analysis

One-way analysis of variance (ANOVA) was employed for analyzing the data and the mean difference between given treatments was intended for significance test at *p* > 0.05 level by Tukey’s HSD. The statistical analysis was performed by using software IBM-SPSS version 25.0.

## 3. Results

### 3.1. Extract Yield (%)

The resultant yield extracted from different parts of *S. marianum* was calculated and presented in (Table 1). The results indicated that the seeds of the *S. marianum* produced a maximum yield of 5.01 % followed by leaves and stem with 3.47 and 2.19%, respectively.

### 3.2. Qualitative Phytochemical Analysis

The analysis of the phytochemicals qualitatively revealed the existence of various phytoconstituents such as alkaloids, glycosides, flavonoids, terpenoids, steroids and catcholic tannins in the seeds, stem and leaves, except for saponins and Gallic tannins. However, phenols were absent in seed extracts, which consequently represent negligible phenolic contents. The results for phytochemicals are represented in (Table 2).

### 3.3. Quantitative Phytochemical Analysis for Total Flavonoids, Phenol, DPPH and FRAP Assay

Contents for total phenols, flavonoids, DPPH scavenging activity and values for FRAP assay demonstrated by extracts of seed, stem and leaves are presented in (Table 3). Results indicated that the highest total phenols were reported from the leaves and stem as 21.79 and 17.29 GAE/g, respectively. On the other hand, the highest total flavonoids were afforded by leaves and stem extracts of 129.66 and 114.29 QE/g followed by seed extracts of 24.72 QE/g, respectively. The results on inhibition % (IC_50_) displayed that all the extracts showed the inhibition of free radicals conferring significant antioxidant activity such as 75.98, 72.39 and 63.21% from the seeds, stem and leaves, respectively. Whereas, the maximum reducing activity was afforded by a stem extract of 51.40 mmol/g followed by seed and leaves extracts of 46.60 and 41.30 mmol/g, respectively.

### 3.4. Chemical Analysis

The analysis for the existence of different chemical compounds in the various parts of *S. marianum* was examined by GC-MS procedures with the presentation of results in (Table 4). Six, eight and ten chemical compounds equivalent to 99.95, 99.96 and 99.89% of the whole extract were identified from seed, stem and leaves, respectively. However, the major chemical compound, Dibutyl phthalate was reported from all extracts while, Hexadecanoic acid, methyl ester and Silane, (1,1-dimethylethyl) dimethyl (phenylmethoxy) were not reported from the stem. Similarly, Benzene, 1-isocyanato-2-methoxy- a major chemical compound was reported from seeds and stem extracts while absent in leaves extract. Moreover, Methyl stearate, also a major compound, was reported from all extracts except for seeds. Similarly, some of the minor compounds were also reported from the extracts having low peak areas.

## 4. Discussion

Natural antioxidants are vital substances, have the ability to avoid oxidative chain reactions and ultimately protect the organism from the adverse effects of toxic radical groups from persuaded oxidative stress. *S. marianum* also holds important flavonoids, which are actually the silymarin compounds famous for curing disorders of the gall bladder and liver in humans [50]. The results obtained from the current study support the opinion of several researchers who predicted the connection of phenolic compounds with antioxidant activity [51,52]. Likewise, Tawaha et al. [53] reported the *S. marianum* lined antioxidant activity relationship with phenolic content. Similarly, various studies on medicinally important plants regarding free radical scavenging activities revealed the efficacy and this activity fluctuates among different plant species. Hadaruga et al. [54] also reported similar activity of essential oil components of *S. marianum*. Moreover, the proximate analysis demonstrated its richness in fats, carbohydrates, proteins, minerals and ash contents.

Furthermore, the contents of silymarin analyzed by HPLC were ranging between 1840.6 and 1765.9 mg/g and 1669.5 and 1607.6 mg/g for microwave-assisted extraction and the soxhlet method, respectively [4] Likewise, the chemical analysis of seed essential oils of *S. marianum* using the GC-MS method showed the existence of 14 volatile components. Among them, the components γ-cadinene (49.8%) and α-pinene (24.5%) were recorded as predominant. Similarly, results showed the prevalence of linoleic and oleic acids as 50.5 and 30.2%, respectively. *S. marianum* exhibited polyphenol contents of 29 mg GAE/g; whereas, tannins and flavonoids were in the range of 1.8 and 3.39 mg EC/g, respectively. However, the foremost recognized phenolic composites were silybin A and B, isosilybin A and B, silychristin, and silydianin 12.2, 17.67, 21.9, 12.8, 7.9, 7.5, respectively [55].

An antioxidant activity assay of in vitro grown tissues of *S. marianum* revealed that a significantly higher antioxidant activity from shoots and seeds was observed as 3.5 and 3.7 mg of fresh tissues [56]. Our findings are consistent with Sun, Li [57] who demonstrated that pappi of *S. marianum* displayed the highest total flavonoid contents from dry plant extract and fruit receptacles at 17.10 mg and 15.34 mg/g, respectively. Similarly, the total phenolic contents ranged from 9.80 to 48.97 mg gallic acid/g of dry plant portion. At 50 µg/mL, the ethanol extract from pappi of *S. marianum* displayed the highest DPPH radical scavenging action 69.68%, and scavenging activity from roots is 66.02% respectively. Qin et al. [58] n isolated Mariamide-A and Mariamide-B (1–2) from *S. marianum* among other 14 known compounds. However, these compounds mostly showed moderate to significant antioxidant activities.

Another study reported that quantification of the yield of plant extracts and essential oil varies within the same plant species, plant parts used for extraction and extraction techniques which also affect chemical composition [59,60]. The findings of the current study are consistent with Akhtar et al. [61] who reported total phenolic contents as 20.2–85.6 mg GAE/g from the methanol extract. Earlier, Akhtar et al. [62] investigated 20.1–25.9% DPPH free radical scavenging activity from methanol and distilled water extract of *S. marianum*, respectively, which are strongly consistent with current findings. Furthermore, the findings of Ismaili et al. [63] are in strong conformity with our findings which reported the antioxidant activity value as 353.89 mg TE/g edw and 8.38 AAE/g edw from Moroccan Milk thistle via FRAP assay, respectively.

On using the GC-MS technique from the essential oil of *Origanum compactum* for nutrient screening and amino acids. The principal component analysis confirmed the presence of carvacrol and thymol as the major constituents, accounting for 14.84 and 78.81% using microwave-assisted extraction and Clevenger hydrodistillation, respectively, consistent with our outcomes [33]. GC-MS analysis of a solvent crude extract of different parts of *S. marianum* demonstrated the six, eight and nine chemical compounds from seed, stem and leaves, respectively, accountable for antioxidant activities. Our findings were supported by Anthony and Saleh [64] who reported seven chemical compounds from the crude extract of *S. marianum* seed including taxifolin, silychristin, silydianin, silybin A, silybin B, iso-silybin A and iso-silybin B. Ningbo et al. [65] isolate mariamides A and B, an amid compound along with 14 identified compounds *S. marianum* seed. Among these isolated compounds most of them showed significant antioxidant activities compared to positive controls except for other amide compounds that showed moderate DPPH radical scavenging activity.

Chemical compounds exhibited by *S. marianum* showed various biological activities, as well significant uses in industries. Methyl stearate is a fatty acid methyl ester used as a nonionic surfactant, to enhance the solubility of different chemicals, a stabilizer and an emulsifier. However, the published literature is limited on Methyl stearate [66]. Dibutyl phthalate is an important plasticizer also used as a solvent for dyes [67]. However, among the pollutants list regulated by the United States Environmental Protection Agency (U.S. EPA), Dibutyl phthalate is one of the six phthalic acid esters causing pollution. Hexadecanoic acid, also known as Palmitic acid, is a common saturated fatty acid exhibited by plants, animals and microbes [68]. It is a chief component of the oil widely used in foodstuffs and a natural additive in organic products [69].

Moreover, the observations of Ningbo et al. [65] also support our findings that the maximum DPPH scavenging activity from the leaves of full-grown *S. marianum* and intact plants was observed at 60 and 65.43%. Similarly, when treated with various gamma radiation dosages it displayed 45.43 and 59.26 DPPH scavenging activity, respectively. Whereas, the plants derived from un-treated gamma radiation, seeds exhibited 44.53% scavenging activity. In a study, seven key components such as silydianin, taxifolin, silychristin, silybin A and B, iso-silybin A and B were identified from the crude extract of *S. marianum* which showed significant free radical scavenging activities; however, among these compounds, taxifolin showed pronounced radical scavenging activity via DPPH assessment bearing an EC_50_ of 32 µM compared to all other components support our findings on antioxidant activities of different parts of *S. marianum* [70]. In a study, it was evaluated that *S. marianum* and *C. siliqua* extracts exhibited scavenging properties on the DPPH free radicals at 94.94 and 93.81%, respectively, [71] which are in agreement with our results. In addition, according to a study, the extract from *S. marianum* possesses high antioxidant activity at 5.77 and 12.14% using TAS (mmol/L) and TOS (µmol/L), respectively [72]. The outcomes of the present work demonstrated that different parts of *S. marianum* possess certain vital phytoconstituents and displayed good antioxidant activities with significant phenols and flavonoids contents. The bioactive components from the extract of *S. marianum* and their use as potential natural antioxidants could be of important economic value. However, supplementary investigations involving more detailed assays on extraction, purification and isolation of biological compounds seem to be needed to determine which components give the best biological activities. Although, some studies had been carried out on the antioxidant activities of this plant, but very few studies have explored the phytochemical analysis qualitatively and quantitatively detailed antioxidant activity, as well as chemical analysis of different portions of *S. marianum* in such a comprehensive way.

## 5. Conclusions

The present study outcomes suggest that *Silybum marianum* contains significant proportions of phytochemicals and an appreciable amount of phenols and flavonoids conferring to the free radical scavenging activity. Moreover, chemical analysis revealed that a significant amount of chemical compounds is present, which causes the antioxidant activity of *S. marianum.* Hence, *S. marianum* could be a contributor to a positive source of treatment for various disorders. However, further studies are desired on the purification and identification of biochemical compounds for the sake of commercial purposes.

## Figures and Tables

**Table 1 molecules-27-02641-t001:** Yield of extracts afforded by different parts of *S. marianum*.

Extracts	Conc. (µg mL^−1^)	Yield (%)
Seed	50	5.01 ± 0.04 ^a^
Stem	50	2.19 ± 0.06 ^c^
Leaves	50	3.47 ± 1.41 ^b^
Statics	S.S	19.83
	M.S	9.92
	Df	4
	*f*	90.40 ***

Data in the columns is demonstrated as mean values; ±standard deviation with various superscripts which are significantly different according to Tukey’s HSD″ *p* > 0.05). M.S (Mean square); S.S (Sum of square); *f* (Significance); *** (level of significance); Df (Degree of freedom). Superscripts a, b, c (Indicate pairwise comparisons whether they are statistically different).

**Table 2 molecules-27-02641-t002:** Qualitative analysis for phytochemicals from different parts of *S. marianum*.

Extracts	Conc. (µg mL^−1^)	Phytoconstituents								
		Alkaloids	Glycosides	Flavonoids	Phenols	Terpenoids	Steroids	Saponins	Gallic Tannins	Catcholic Tannins
Seed	50	+	+	+	-	+	+	-	-	+
Stem	50	+	+	+	+	+	+	-	-	+
Leaves	50	+	+	+	+	+	+	-	-	+

+ (presence of phytoconstituents); - (absence of phytoconstituents).

**Table 3 molecules-27-02641-t003:** Total phenols, flavonoids content and DPPH inhibition % of *S. marianum*.

Extracts	Conc. (µg mL^−1^)	Total Phenols (GAE/g)	Total Flavonoids (QE/g)	DDPH Inhibition (IC_50_)	FRAP Assay mmol/g
Seed	50	1.70 ± 0.03 ^c^	24.72 ± 0.39 ^b^	75.98 ± 0.14 ^a^	46.60 ± 0.51 ^a^
Stem	50	17.29 ± 0.03 ^b^	114.29 ± 0.20 ^c^	72.39 ± 0.49 ^b^	51.40 ± 0.51 ^b^
Leaves	50	21.79 ± 0.18 ^a^	129.66 ± 0.65 ^a^	63.21 ± 0.16 ^c^	41.30 ± 0.49 ^c^

Data in the columns is demonstrated as mean values ± standard deviation with various superscripts which are significantly different according to Tukey’s HSD″ *p* > 0.05). Superscripts a, b, c (Indicate pairwise comparisons whether they are statistically different).

**Table 4 molecules-27-02641-t004:** Chemical composition of seeds, stem and leaves extract of *S. marianum*.

Extract	PK #	RT	Area %	Chemical Compounds	M.F g/moL	M. Wt.
Seed	1	3.38	19.72	Benzene, 1-isocyanato-2-methoxy-	C_8_H_7_NO_2_	149.1
	2	3.75	11.02	Silane, (1,1-dimethylethyl)dimethyl(phenylmethoxy)-	C_13_H_22_OSi	222.4
	3	40.41	35.55	Hexadecanoic acid, methyl ester	C_18_H_36_O	268.5
	4	41.65	10.48	Dibutyl phthalate	C_16_H_22_O_4_	278.3
	5	45.87	13.72	8-Octadecenoicacid, methyl ester	C_19_H_36_O_2_	296.5
	6	46.68	9.46	Heptadecanoic acid, 14-methyl-, methyl ester, (.+/-.)-	C_18_H_36_O_2_	284.4
Stem	1	3.42	8.18	Benzene, 1-isocyanato-3-methoxy-	C_8_H_7_NO_2_	149.1
	2	7.79	7.98	Carbamic acid, (3-methylphenyl)-, methyl ester	C_9_H_11_NO_2_	165.2
	3	37.72	4.21	2-Pentadecanone, 6,10,14-trimethyl	C_18_H_36_O	268.5
	4	38.56	31.84	1,2-Benzenedicorboxylic acid, butyl decyl ester	C_22_H_34_O_4_	362.50
	5	40.41	26.01	Pentadecanoic acid, 14-methyl-, methyl ester	C_17_H_34_O_2_	270.5
	6	41.66	9.04	Dibutyl phthalate	C_16_H_22_O_4_	278.34
	7	45.87	6.53	11-Octadecenoic acid, methyl ester	C_19_H_36_O_2_	296.5
	8	46.69	6.20	Methyl stearate	C_19_H_38_O_2_	298.5
Leaves	1	2.72	6.18	2-(2-Methoxy-5-methyl-phenyl)-propionaldehyd	C_11_H_14_O_2_	178.2
	2	3.79	8.98	Silane, (1,1-dimethylethyl)dimethyl (phenylmethoxy)-	C_9_H_10_O_3_	166.2
	3	27.72	3.62	2-Undecanone, 6,10-dimethyl-	C_13_H_26_O	198.3
	4	34.15	28.84	2-Pentadecanone, 6,10,14-trimethyl-	C_16_H_22_O_4_	278.34
	5	39.87	6.94	7-Octadecenoic acid, methyl ester	C_19_H_36_O_2_	296.5
	6	40.47	8.04	Hexadecanoic acid, methyl ester	C_13_H_22_OSi	222.4
	7	41.62	25.08	Dibutyl phthalate	C_18_H_36_O	268.5
	8	46.69	7.54	Methyl stearate	C_19_H_38_O_2_	298.5
	9	47.12	3.67	Tetradecanoic acid, 12-methyl-, methyl ester	C_16_H_32_O_2_	256.4

M.F (Molecular Formula); M.W (Molecular Weight); R.T (Retention time).

## Data Availability

All the data is available in the manuscript file.

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
