# Peer review of "Comparative Assessment of Phytoconstituents, Antioxidant Activity and Chemical Analysis of Different Parts of Milk Thistle Silybum marianum L."

_molecules, 2022, doi:10.3390/molecules27092641_

Round 1

Reviewer 1 Report

You did not take into account many of my comments (detailed descriptions of the methods are still missing, e.g., what was the ratio of  between methanol and sample during the cold extraction process, what was the control for DPPH made of, how was prepared the blank for flavonoids , what was the reference for DPPH method  etc.).
The text lacks a reasonable explanation why the retention times for the same compound of different origin differ and moreover, how is it possible that the same compound eluted from the column twice (as determined by GC-MS?! dibutyl phthalate extracted from a) seeds has had Rt 41.65 min, b)from steam 38.65 and 41.67 and c)from leaves 34.15 (Table 4).
You also did not improve their discussion according to my comments made earlier (1st round of revision).

Author Response

Response to Reviewer. 1

Dear Sir/Madam, thank you very much for giving us a chance again to polish our work with your kind comments and suggestions on our article, which helped us to improve the quality of our article. We have followed your comments & suggestions and made improvements as follows:

Q: What was the ratio of between methanol and sample during the cold extraction process?

Reply: For extraction, 10 g of each dried sample was extracted with 100 mL of methanol 95% separately with ratio (10:100).

Q: What was the control for DPPH?

Reply:  Methanol was used as a negative control, whereas, DPPH and methanol was used for experimental control

Q: How was prepared the blank for flavonoids?

Reply: The blank was executed using distilled water.  

Q: What was the reference for DPPH method etc.?

Reply: Three references were cited for DPPH assay, (Yu et al., 2003; Zhao et al., 2008; Vijayalakshmi, et al. 2013)
Q: The text lacks a reasonable explanation why the retention times for the same compound of different origin differ and moreover, how is it possible that the same compound eluted from the column twice (as determined by GC-MS?! dibutyl phthalate extracted from a) seeds has had Rt 41.65 min, b)from steam 38.65 and 41.67 and c)from leaves 34.15 (Table 4).

Reply: Compliance was done. After go through the mother file obtained by GC-MS analysis and obtained Chromatogram, The results were revised and added in the Table 4. The retention time was also revised and corrected according to results as obtained by GC-MS analysis. The retention time for Dibutyl phthalate extracted from seeds, stem and leaves was 41.65, 41.66 and 41.62, respectively, revised and corrected (Table 4) as suggested by the reviewer.

Q: You also did not improve their discussion according to my comments made earlier (1st round of revision).

Reply: Discussion section was revised and improved. More supporting statements and references were added regarding chemical compounds analyzed by GC-MS analysis and industrial and pharmacological use of extracted chemical compounds.  However, the retention time may differ because of the quality of the phytoconstituents exhibited by different by different parts of the plant.

Reviewer 2 Report

Dear authors, 

Your submission contains interests results. However, it needs a major revision to be recommended for publication in Molecules. Please refer to my comments and suggestions in the attached file. 

Regards. 

Author Response

Response to Reviewer. 2

Dear Sir/Madam, thank you very much for giving us a chance again to polish our work with your kind comments and suggestions on our article, which helped us to improve the quality of our article. We have followed your comments & suggestions and made improvements as follows:

Reviewer’s suggestions: Your submission contains interest’s results. However, it needs a major revision to be recommended for publication in Molecules. Please refer to my comments and suggestions in the attached file. 

Compliance has been done as, grammatical mistakes have been corrected, and suggestion were followed and revised and corrected in the manuscript file. Moreover, question were extracted from the pdf file of the manuscript and reply were given as, Introduction part was revised and three citation were added as suggested by the reviewer. 

Q 1:

Reply: Manuscript has revised thoroughly

Q 2:

Reply: statement was checked and revised regarding grammar

Q3:

Reply: Instruction were followed throughout the manuscript

Q 4:

Reply: suggested statement was re-phrased

Q 5:

Reply: Suggested references has been added and cited in the manuscript.

Q 6:

Reply: Section 2.1 was more elaborated

Q 7:

Reply:  methanol was selected because of the following reasons

Being a polar solvent, methanol has good penetration to the cell content and exhibited the capacity to extract both the polar and non-polar compounds and to dissolve primary and secondary metabolites. Moreover, methanol boils at 64.7°C and temperature needs to lower down to evaporate the solvent which in return less damages the extract. For extraction, 10 g of each dried sample was extracted with 100 mL of methanol 95% separately with sample to solvent ratio (10:100) (Li P, Xu G, Li SP, 2008; Li P, Yin ZQ, Li SL, 2014). Three replication were performed for each sample extraction.

Q 8:

Reply: Equation was revised as

Q 9:

Reply: Total essential oil was revised as “total extract”

Q 10:

Reply:  Table 2 was adjusted as suggested by the reviewer…

Q 11:

Reply:  Table 4, 5, 6 were merged as suggested by the reviewer.

Reviewer 3 Report

The authors reply to all my queries.

Author Response

Response to Reviewer. 3

Dear Sir/Madam, thank you very much for giving us a chance to polish our work with your kind comments and suggestions on our article, which helped us to improve the quality of our article. We have followed your comments & suggestions and made improvements.

As your comments indicated that we have addressed all your recommendations, therefore we are thankful to you.

Round 2

Reviewer 1 Report

Thank you for all your answers. You have successfully upgraded your manuscript. I don't have any additional comments.

Reviewer 2 Report

Dear authors, 

It seems that the manuscript has been greatly improved and therefore I recommend its publication in Molecules. 

Regards. 

This manuscript is a resubmission of an earlier submission. The following is a list of the peer review reports and author responses from that submission.

Round 1

Reviewer 1 Report

Language needs improvement.
Introduction: is adequate.
Materials and methods: they are deficient, it is necessary to give more information about the material itself, the age, the maturity of the plants, the amount of samples (plants). There is a lack of information about the state of storage of the samples, how long and under what conditions you made extracts from them.
Write in how many replicates you performed the extraction, describe the extraction conditions in detail (e.g. solvent-sample ratio, describe the extraction conditions in detail). All methods must have an appropriate reference, for spectrophotometric methods write down what the blank was. Explain equation 1 in more detail (what the theoretical and actual yield mean).
Results:
Section 3.3, I do not understand your last statement because the results in Table 3 do not match it (DPPH and FRAP order).
Table 5: what is the difference between compound 4 and 10?
Discussion: is quite flawed, not quality enough to for the article to be accepted as is. In your investigation (as evident from the title) you focus on the comparison between different parts of the plant, it lacks a discussion of these results, explain what you think are the reasons for the differences between them. Moreover, it is not enough to mention the results of other researches, it is necessary to link them meaningfully with yours (e.g. you have given information about how many phenols, flavonoids, antioxidant activity others have determined for S. marianum, but not (in most cases) from which parts of the plant they have prepared extracts and under which conditions. Also, you decided to determine the chemical composition of various plant parts (Tables 4, 5, 5), in the discussion we learn nothing about these compounds, why they are important, if at all, and how do these compounds contribute to antioxidant activity as you claim in the conclusion? Manuscript needs to be expanded and improved.

Reviewer 2 Report

Dear Authors, 

Your manuscript contains important results. I see that major changes are needed for its recommendation for the publication in Molecules. Please find my comments and suggestions are embedded in the attached file. 

Regards. 

Reviewer 3 Report

Dear editor, I have reviewed the manuscript, and it should not be published, in my opinion.

There are several flaws:

  • the abstract exceeds the maximum length
  • the introduction is very short and it does not give sufficient background for the study
  • "at Biopesticides lab of the College of Plant Protection,
    Shenyang Agricultural University, China in 2019." this is to avoid in a scientific paper
  • the material and methods section has very limited references (section 2.2, 2.4, 2.5, 2.6, 2.7 NO ONE)
  • the sampling (section 2.1) is very poor and it does not give some crucial information
  • section 2.2 is really suitable this kind of extraction for a GC analysis? 
  • Table 2 is very hard to read
  • strange notation in table 3 (GAE^-g), never seen
  • what is the meaning of including the static summary in table 3, since they were not discussed?
  • "The results obtained from the current study are supporting the opinion of several researchers who predicted the connection of phenolic compounds with antioxidant activity". The authors don't perform a correlation
  • The discussion section is really confused and it seems that the authors are not discussing their results
  • The conclusions section is very poor